# Effectiveness of Naldemedine Compared with Magnesium Oxide in Preventing Opioid-Induced Constipation: A Randomized Controlled Trial

**DOI:** 10.3390/cancers14092112

**Published:** 2022-04-24

**Authors:** Anna Ozaki, Takaomi Kessoku, Kosuke Tanaka, Atsushi Yamamoto, Kota Takahashi, Yuma Takeda, Yuki Kasai, Michihiro Iwaki, Takashi Kobayashi, Tsutomu Yoshihara, Takayuki Kato, Akihiro Suzuki, Yasushi Honda, Yuji Ogawa, Akiko Fuyuki, Kento Imajo, Takuma Higurashi, Masato Yoneda, Masataka Taguri, Hiroto Ishiki, Noritoshi Kobayashi, Satoru Saito, Yasushi Ichikawa, Atsushi Nakajima

**Affiliations:** 1Department of Gastroenterology and Hepatology, Yokohama City University Graduate School of Medicine, Yokohama 236-0004, Japan; anx0513ro@hotmail.com (A.O.); kosuke.tsssik@gmail.com (K.T.); atsushi.y.0410@gmail.com (A.Y.); takahashi1700pk9@gmail.com (K.T.); y.kasai.91@gmail.com (Y.K.); michihir@yokohama-cu.ac.jp (M.I.); tkhkcb@gmail.com (T.K.); t_yoshi@yokohama-cu.ac.jp (T.Y.); rainbowman0803@gmail.com (Y.H.); yuji.ogawa01@gmail.com (Y.O.); fuyukia@yokohama-cu.ac.jp (A.F.); kento318@yokohama-cu.ac.jp (K.I.); takuma_h@yokohama-cu.ac.jp (T.H.); yoneda@yokohama-cu.ac.jp (M.Y.); ssai1423@yokohama-cu.ac.jp (S.S.); nakajima-tky@umin.ac.jp (A.N.); 2Department of Palliative Medicine, Yokohama City University Hospital, Yokohama 236-0004, Japan; ytd0714@gmail.com (Y.T.); yasu0514@yokohama-cu.ac.jp (Y.I.); 3Department of Gastroenterology, International University of Health and Welfare Atami Hospital, Atami 413-0012, Japan; t.kato222@gmail.com; 4Department of Oncology, Yokohama City University Hospital, Yokohama 236-0004, Japan; akihiroweapon@yahoo.co.jp (A.S.); norikoba@yokohama-cu.ac.jp (N.K.); 5Department of Data Science, Yokohama City University Graduate School of Medicine, Yokohama 236-0004, Japan; taguri@yokohama-cu.ac.jp; 6Department of Palliative Medicine, National Cancer Center Hospital, Tokyo 104-0045, Japan; ishiki-tky@umin.ac.jp

**Keywords:** opioid-induced constipation, magnesium oxide, naldemedine, spontaneous bowel movement

## Abstract

**Simple Summary:**

Opioids are used in cancer pain management, however, their continuous use may not be tolerable owing to adverse effects such as constipation, sleepiness, nausea, and respiratory depression. Opioid-induced constipation reduces the quality of life of patients, and osmotic laxatives are conventionally recommended for preventing opioid-induced constipation. Recently, naldemedine, a peripherally acting μ-opioid receptor antagonist, can be used to safely and effectively treat opioid-induced constipation based on its etiological mechanism, without affecting central analgesia. In this study, we compared the effectiveness of magnesium oxide with that of naldemedine in preventing opioid-induced constipation. Naldemedine significantly prevented deterioration in the quality of defecation (the Japanese Patient Assessment of Constipation Quality of Life and complete spontaneous bowel movement) and reduced gastrointestinal adverse effects, mainly nausea, compared with magnesium oxide during 12-week administration.

**Abstract:**

Opioid-induced constipation (OIC) may occur in patients receiving opioid treatment, decreasing their quality of life (QOL). We compared the effectiveness of magnesium oxide (MgO) with that of naldemedine (NAL) in preventing OIC. This proof-of-concept, randomized controlled trial (registration number UMIN000031891) involved 120 patients with cancer scheduled to receive opioid therapy. The patients were randomly assigned and stratified by age and sex to receive MgO (500 mg, thrice daily) or NAL (0.2 mg, once daily) for 12 weeks. The change in the average Japanese version of Patient Assessment of Constipation QOL (JPAC-QOL) from baseline to 2 weeks was assessed as the primary endpoint. The other endpoints were spontaneous bowel movements (SBMs) and complete SBMs (CSBMs). Deterioration in the mean JPAC-QOL was significantly lower in the NAL group than in the MgO group after 2 weeks. There were fewer adverse events in the NAL group than in the MgO group. Neither significant differences in the change in SBMs between the groups nor serious adverse events/deaths were observed. The CSBM rate was higher in the NAL group than in the MgO group at 2 and 12 weeks. In conclusion, NAL significantly prevented deterioration in constipation-specific QOL and CSBM rate compared with MgO.

## 1. Introduction

Opioids are used in cancer pain management [1,2], however, their continuous use may not be tolerable owing to adverse effects such as constipation, sleepiness, nausea, and respiratory depression [3,4,5,6]. Reportedly, 15–64% of patients receiving strong opioid analgesic treatment experience constipation [7,8,9,10,11], and the cumulative incidence of opioid-induced constipation (OIC) in Japan has been reported to be as high as 79%, which has been observed in patients with breast cancer [12]. Prolonged opioid administration is largely associated with OIC [13] and the prophylactic administration of laxatives is important, as drug tolerance in patients with OIC is low [14]. OIC is worth investigating as the symptoms associated with constipation (abdominal pain, bloating, and appetite loss) may reduce the quality of life (QOL) of these patients.

Conventional OIC treatments include non-drug therapy such as the consumption of a fiber-rich diet and use of medications such as laxatives. In Japan, osmotic laxatives are recommended for OIC treatment [15], and an observational study in Japan revealed that prophylactic magnesium oxide intake at the start of opioid therapy attenuated OIC [16]. Therefore, osmotic laxatives, including magnesium oxide, have been used in Japan for the treatment of OIC caused by opioids that act on μ-receptors in the enteric nerves and impair intestinal motility and secretion [6,17]. Additionally, other osmotic laxatives such as polyethylene glycol are used for treating and as the prophylaxis of OIC in many countries. However, neither diet therapy nor osmotic laxative treatment targets the etiological mechanism of OIC [3,9].

Patients with OIC may feel frustrated, stressed, or anxious because of their dietary restrictions and may feel embarrassed for taking frequent and prolonged bathroom breaks. OIC reduces the QOL of patients and requires preventive treatment. Although there has been progress in research on OIC treatment [9], naldemedine, a peripherally acting μ-opioid receptor antagonist (PAMORA), can be used to safely and effectively treat OIC [18,19] based on its etiological mechanism, without affecting central analgesia [20]. In this study, we compared the effectiveness of magnesium oxide with that of naldemedine in preventing OIC.

## 2. Materials and Methods

### 2.1. Study Design

We conducted a single-center, open-label, two-arm, phase II randomized controlled trial between 26 March 2018 and 30 June 2019 in Yokohama City University Hospital. We included 120 adult patients with any type of cancer (aged 20–85 years) who were scheduled to start opioid therapy for cancer pain. The participants were capable of oral intake and providing written consent to participate in the study; they were expected to remain in stable pathological condition during the observation period. Detailed inclusion and exclusion criteria are shown in Appendix A. The study design is outlined in Figure 1. This study was conducted in accordance with the Declaration of Helsinki and approved by the Ethics Committee of Yokohama City University Hospital (approval number: B180301006, approval date: 22 March 2018) prior to the initiation of the study. This trial was registered at the University Hospital Medical Information Network (UMIN) Clinical Trials Registry (UMIN000031891 on 25 March 2018). All patients provided written informed consent. The trial protocol was described according to the standard protocol items: Recommendations for Interventional Trials Patient-Reported Outcome Extension and its checklists (Appendix A) [21]. The results of this trial were reported in conformity with the Consolidated Standards of Reporting Trials 2010 guidelines [22].

### 2.2. Randomization and Masking

The patients were randomly allocated (1:1 ratio) using a computer-based system and stratified by age and sex to the magnesium oxide group (MgO group; 500 mg, thrice daily after each meal) or the naldemedine group (NAL group; 0.2 mg, once daily after breakfast), and each drug was administrated orally for 12 weeks. Randomization was performed using a computer-generated, centrally administered procedure and a permuted block method. Randomization was conducted independently using a validated allocation system (International University of Health and Welfare Atami Hospital, Japan, performed by T. Kato).

### 2.3. Endpoints

The summary of endpoints is shown in Appendix A. The primary endpoint was the change in the Japanese Patient Assessment of Constipation Quality of Life (JPAC-QOL) score from baseline to 2 weeks after treatment initiation. The primary endpoint was calculated from the mean of the difference from baseline at 2 weeks, as opioid-induced constipation develops within 2 weeks of opioid treatment; hence, the primary endpoint was set to 2 weeks. The JPAC-QOL is a reliable and valid psychometric evaluation criterion for patients with functional constipation [23] and comprises 28 questions rated on a five-point adjective score from 0 to 4. A lower the score indicates a higher QOL [24,25,26].

The secondary endpoints were the change in the JPAC-QOL score from baseline to 12 weeks and the changes in the Patient Assessment of Constipation Symptoms (PAC-SYM) [27]; spontaneous bowel movements (SBMs); Bristol stool form scale (BSFS) [28]; constipation scoring system (CSS) [29]; Rome IV [30]; and short form-36 (SF-36) scores [31] at 2 and 12 weeks after treatment initiation. The changes in complete spontaneous bowel movement (CSBM), JPAC-QOL and JPAC-SYM subscales, and numerical rating score (NRS) for pain were assessed using a post hoc analysis. SBM was defined as the number of defecations not induced by rescue medication. CSBM was defined as the number of defecations not induced by rescue medication and not accompanied by a sense of incomplete evacuation [32], indicating the patient’s greater QOL at defecation. According to the European Medicines Agency guidelines, patient CSBM is important, as it incorporates spontaneity and completeness [33].

### 2.4. Statistical Analysis

A retrospective analysis of magnesium oxide/naldemedine in 10 OIC patients in Yokohama City University Hospital showed a mean JPAC-QOL change of −1.19 and −0.76 in the NAL and MgO groups, respectively. We decided to calculate the appropriate number of patients required for a proper analysis of the variance F-test based on these data. Assuming mean changes in the JPAC-QOL score in the NAL group and the MgO group to be −1.19 and −0.76, respectively, with a common standard deviation of 0.76, 51 patients were needed in each group to reach 90% statistical power with a two-sided significance level of 5%. To compensate for any dropout, we proposed increasing the number of patients to 60 per group. To reach this number, 120 patients were needed.

The intention-to-treat (ITT) population, which included all patients who underwent randomization, was used to assess the primary efficacy endpoint, which was set as the change in the mean JPAC-QOL score between weeks 0 and 2. The primary efficacy analysis was performed on the ITT population. The primary endpoint was a continuous variable and was performed using the Student’s *t*-test to compare the two groups. Two-tailed *p* < 0.05 indicated statistical significance. Secondary and tertiary endpoints were analyzed similarly. Chi-squared tests were used to assess the categorical variables, such as the frequency of constipation filling the ROME IV criteria and AEs. The intensity of an AE was graded according to the National Cancer Institute’s Common Terminology Criteria for Adverse Events (NCI-CTCAE) version 5.0. Safety and tolerability analyses were performed on the safety population, which included all patients who received at least one dose of the study drug. JMP software version 11.2.0 (SAS Institute, Cary, NC, USA) was used for all statistical analyses. This study was overseen by an independent medical monitor (on-site monitoring).

## 3. Results

### 3.1. Baseline Characteristics

Of the 166 patients eligible for this study, 120 were included (Figure 1). There was no significant difference in the baseline characteristics of patients in the MgO and NAL groups (Table 1). The MgO group comprised 23 males (38%) and 37 females (62%) (51 ± 9 years old), whereas the NAL group comprised 24 males (40%) and 36 females (60%) (52 ± 9 years old). The Eastern Cooperative Oncology Group (ECOG) performance status (PS) was 0–2 in 54 patients (90%) in the MgO group and in 52 patients (87%) in the NAL group. None of the patients in either group received chemotherapy within 2 weeks of the baseline. Between 2 and 12 weeks, 27 (45%) patients in the MgO group and 27 (45%) patients in the NAL group received chemotherapy. Moreover, in both the MgO and NAL groups, the use of platinum agents (cisplatin, carboplatin, and oxaliplatin) was 37%; the use of taxane agents (paclitaxel) was 22% and 15%, respectively; the use of anti-metabolite agents (tegafur/gimeracil/oteracil, and fluorouracil) was 37% and 48%, respectively; no irinotecan was used in either group.

Five patients (9%) in the MgO group and five (8%) in the NAL group used regular stimulant laxatives; 17 patients (28%) in the MgO group and 13 (22%) in the NAL group used rescue stimulant laxatives; 38 patients (63%) in the MgO group and 42 (70%) in the NAL group did not use any laxatives. Furthermore, strong opioid use was reported in 45% and 49% of patients in the MgO and NAL groups, respectively. The average oral morphine-equivalent opioid dose in the MgO and NAL groups was 13 mg at baseline, 14 mg and 13 mg at 2 weeks, and 22 mg and 23 mg at 12 weeks.

The patients’ mean JPAC-QOL at baseline in both groups was 0.9, whereas the number of SBMs per week was 4.3 and 4.5 and the number of CSBMs per week was 3.8 and 3.7 in the MgO and NAL groups, respectively. The average stool consistency score based on the BSFS scale was 3.8 in the MgO group and 3.6 in the NAL group at baseline.

### 3.2. Primary and Secondary Endpoints

After administration, the change in the overall mean JPAC-QOL from baseline was +0.5 in the MgO group and −0.01 in the NAL group (*p* < 0.001, lower scores indicate greater QOL) at 2 weeks, and +0.4 in the MgO group and +0.03 in the NAL group (*p* < 0.001) at 12 weeks (Figure 2A). The change in the overall mean PAC-SYM from baseline was +0.6 in the MgO group and +0.02 in the NAL group (*p* < 0.001) at 2 weeks, and +0.5 in the MgO group and +0.02 in the NAL group (*p* < 0.001) at 12 weeks (Figure 2B). There was no difference in the frequency of SBMs between the groups (Figure 3A). However, the post hoc analysis revealed higher mean CSBMs in the NAL group than in the MgO group at both 2 and 12 weeks (0 vs. −0.9, *p* = 0.01 at 2 weeks and +0.2 vs. −0.7, *p* = 0.003 at 12 weeks) (Figure 3B). The PAC-QOL subscale (physical discomfort, psychosocial discomfort, and satisfaction) and all mean PAC-SYM subscale (stool symptoms, rectal symptoms, and abdominal symptoms) scores were significantly improved at 2 and 12 weeks (Table 2 and Figure 4A–D). After 2 and 12 weeks of treatment, the number of patients diagnosed with constipation by Rome IV criteria was significantly lower in the NAL group (Table 2). All SF-36 subscales, including the physical, mental, and role component summary, showed no significant differences between the groups at 2 and 12 weeks (Table 2). There was no significant difference in the average time to the first SBM, but the mean time to the first CSBM was significantly shorter in the NAL group at 2 weeks (10.4 h vs. 6.4 h; *p* < 0.001) and 12 weeks (10.1 h vs. 6.4 h; *p* < 0.001) (Table 2). No significant change in the NRS was observed at 2 and 12 weeks (Table 2).

### 3.3. Safety Outcomes

AEs are listed in Table 3. The rate of treatment-related AEs (TRAE) was significantly higher in the MgO group than in the NAL group (MgO; 35% vs. NAL; 18%, *p* = 0.02). TRAEs were observed in 21 patients in the MgO group and in 11 patients in the NAL group. Nausea accounted for the highest proportion of TRAEs in the MgO group at 2 weeks. At 12 weeks, the rate of TRAEs was significantly higher in the MgO group than in the NAL group (MgO; 52% vs. NAL; 27%, *p* = 0.02). TRAEs occurred in 31 patients in the MgO group and in 16 patients in the NAL group. Nausea also accounted for the highest proportion of TRAEs in the MgO group at 12 weeks. No serious AEs or death occurred during the study period.

## 4. Discussion

This proof-of-concept, two-arm, phase II clinical trial demonstrated that the deterioration in JPAC-QOL was significantly lower in the NAL group than in the MgO group after 2 and 12 weeks of drug administration. Therefore, our trial met the primary endpoint. Higher CSBM rates were calculated in the NAL group at 2 and 12 weeks via post hoc analysis. However, no difference in defecation frequency (SBMs) between the MgO and NAL groups at 2 and 12 weeks was observed. Patients in both groups experienced >3 SBMs/week (<3 SBMs/week is one criterion defining constipation in Rome IV).

Several confounding factors were considered for endpoint evaluation, particularly defecation, food intake [34], decreased physical function (frailty) [35], and medication such as opioid [36] and chemotherapy [37], which are associated with gastrointestinal symptoms such as constipation and diarrhea. Opioid-induced constipation is exacerbated as opioid doses increase [36]. However, our results show no significant difference between the groups in terms of oral morphine-equivalent daily dose at baseline and at 2- and 12-weeks post-treatment. Therefore, it is unlikely that the opioids increased the potential risk of OIC in our study.

In addition, the type of chemotherapy [38,39], taxane agents [40], anti-metabolite agents [41], irinotecan [42], and antiemetics (mainly 5-hydroxytryptamine 3 receptor and neurokinin 1 receptor antagonists [43,44,45]) used for treatment are other confounding factors in constipation and diarrhea. However, as shown in Table 1, no significant differences were observed between the groups, which suggests that the effect of these factors on the primary endpoint is negligible.

PAMORAs, including NAL, are used for OIC treatment and are currently recommended in cases where osmotic or stimulant laxatives are ineffective [46]. However, at present, its long-term effects remain unknown. This study illustrated the efficacy and safety of the prophylactic effect on OIC for up to 12 weeks. Additionally, fewer AEs, especially opioid-induced nausea and vomiting (OINV), were observed in the NAL group at both 2 and 12 weeks. In an animal model, Kanemasa et al. reported the antiemetic properties of naldemedine and its efficacy against OINV [47]. The secondary effects of naldemedine on OINV have been reported by Sato et al., who showed that using naldemedine at an early stage of opioid administration may have secondary benefits in patients with constipation, such as relief from OINV, in addition to improving OIC [48]. OINV occurs when opioids stimulate the peripheral μ-opioid receptors, thereby altering gastrointestinal motility and function [14], which may be prevented as naldemedine antagonizes these receptors.

This study indicates that magnesium oxide or naldemedine may be used to prevent OIC, although naldemedine significantly prevented deterioration in constipation-specific QOL and CSBM compared with magnesium oxide. One of the advantages of magnesium oxide is that its long-term safety has been empirically established through the conventional use of magnesium oxide for OIC prevention in Japan. Additionally, magnesium oxide is cost-effective, as it costs USD 0.3 (JPY 33.6) per day for a dose of 1500 mg/day whereas naldemedine costs USD 2.6 (JPY 272.1) per day.

There were some limitations to our study. This was a single-center, open-labeled study, and the treatment period (12 weeks) may have been too short to investigate the long-term effects. Therefore, large-scale multicenter blind studies with long-term follow-ups are warranted.

## 5. Conclusions

When treating OIC in patients with cancer, naldemedine significantly prevented deterioration in constipation-specific QOL and CSBM compared with magnesium oxide. Future studies should evaluate the clinical benefits of naldemedine over magnesium oxide, keeping in mind its cost.

## Figures and Tables

**Figure 1 cancers-14-02112-f001:**
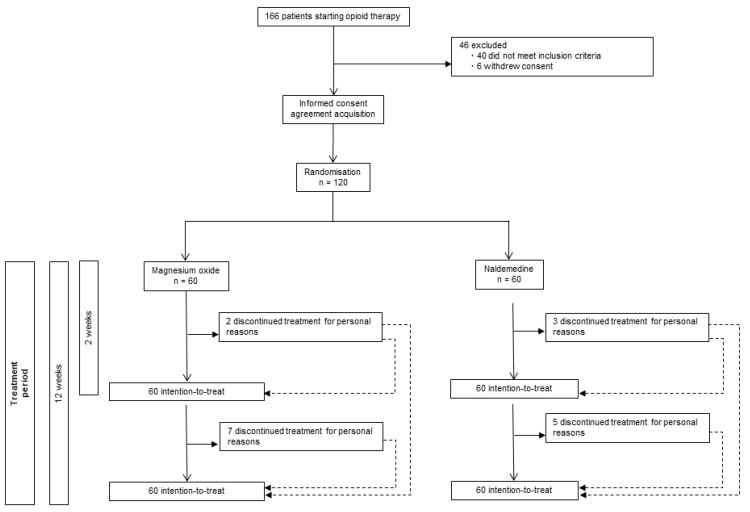
Flow chart showing the study outline.

**Figure 2 cancers-14-02112-f002:**
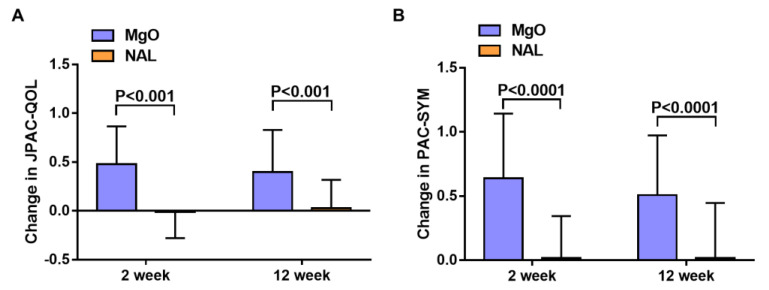
(**A**): Overall Japanese version of Patient Assessment of Constipation Quality of Life (JPAC-QOL) score at baseline and after 2 and 12 weeks of treatment in the magnesium oxide group and naldemedine groups; and (**B**) overall Patient Assessment of Symptoms (PAC-SYM) score at baseline and after 2 and 12 weeks of treatment in the magnesium oxide group and naldemedine groups.

**Figure 3 cancers-14-02112-f003:**
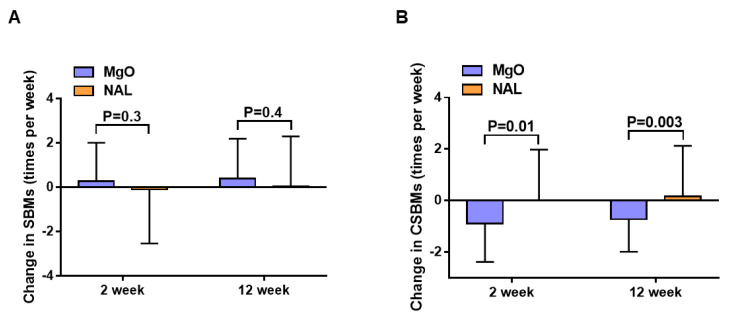
(**A**) Change in the number of spontaneous bowel movements (SBMs; times/week) from baseline to 2 and 12 weeks (after treatment) in the magnesium oxide group and the naldemedine group; and (**B**) change in the number of complete spontaneous bowel movements (CSBMs; times/week) from baseline to 2 and 12 weeks (after treatment) in the magnesium oxide group and the naldemedine group.

**Figure 4 cancers-14-02112-f004:**
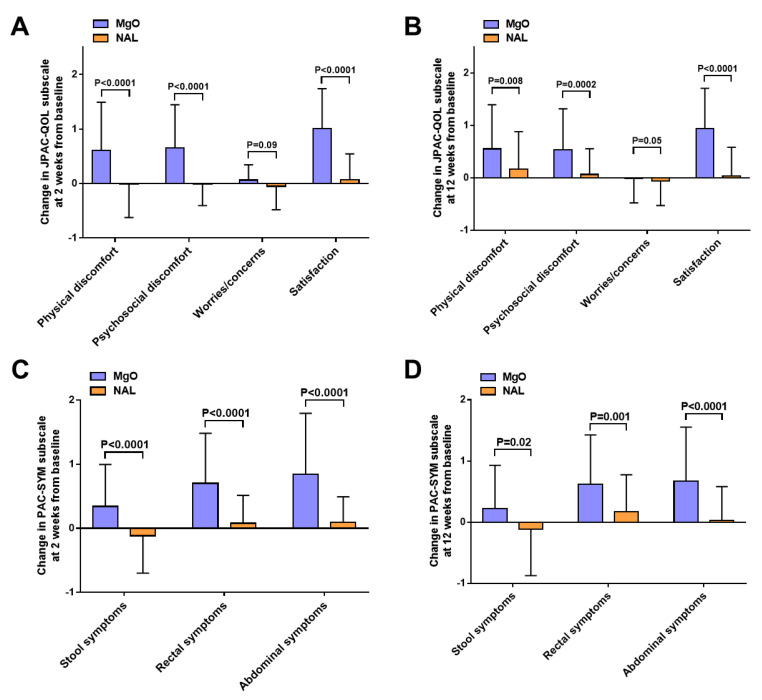
(**A**) Change in the Japanese version of Patient Assessment of Constipation Quality of Life (JPAC-QOL) subscale score at baseline and after 2 weeks of treatment in the magnesium oxide (MgO) group and naldemedine (NAL) groups; (**B**) change in the JPAC-QOL subscale score at baseline and after 12 weeks of treatment in the MgO group and NAL groups; (**C**) change in the Patient Assessment of Constipation Symptoms (PAC-SYM) subscale score at baseline and after 2 weeks of treatment in the MgO group and NAL groups; and (**D**) change in the PAC-SYM subscale score at baseline and after 12 weeks of treatment in the MgO group and NAL groups.

**Table 1 cancers-14-02112-t001:** Demographic and baseline characteristics of modified intention-to-treat populations.

Characteristic	1500 mg MgO	0.2 mg NAL
	(*n* = 60)	(*n* = 60)
Age (years)	51 (9)	52 (9)
Sex		
Female	37 (62)	36 (60)
Male	23 (38)	24 (40)
Body mass index (kg/m^2^)	22 (4)	22 (10)
History of abdominal operation	33 (55)	33 (55)
ECOG PS 0–2	54 (90)	52 (87)
Palliative prognosis index ≤ 3.5	56 (93)	53 (88)
Primary tumor site		
Hepatobiliary and pancreas	18 (30)	21 (35)
Gastrointestinal tract	13 (22)	16 (27)
Lung	5 (8)	3 (5)
Others	24 (40)	20 (33)
Concurrent cancer treatment		
Chemotherapy (0–14 days)	0	0
Chemotherapy (15–84 days)	27 (45)	27 (45)
Chemotherapy type		
Platinum agents	10 (37)	10 (37)
Taxane agents	6 (22)	4 (15)
Anti-metabolite agents	10 (37)	13 (48)
Irinotecan	0	0
Antiemetics during chemotherapy ^#^	3 (11)	2 (7)
Perioperative	12 (20)	12 (20)
Best supportive care	9 (15)	14 (23)
Others	12 (20)	7 (12)
Concomitant medications		
Laxative use		
Naïve	38 (63)	42 (70)
Regular use (irritant laxative)	5 (8)	5 (8)
Rescue use (irritant laxative)	17 (28)	13 (22)
Opioid use at baseline		
Strong opioid	27 (45)	30 (50)
Weak opioid	33 (55)	30 (50)
Mean total daily dose of opioid *		
At baseline (mg)	13 (4)	13 (4)
At 2 weeks (mg)	14 (4)	13 (5)
At 12 weeks (mg)	22 (19)	23 (23)
Baseline defecation status		
Mean JPAC-QOL	0.9 (0.6)	0.9 (0.4)
SBMs per week	4.3 (1.7)	4.5 (2.8)
CSBMs per week	3.8 (1.5)	3.7 (2.1)
Stool consistency score	3.8 (0.7)	3.6 (1.9)

Data are represented as the mean (SD) or number (%). Age was based on the date of informed consent. Baseline values were based on the last week before the start of drug administration. Stool consistency was assessed according to the Bristol stool form scale scores. * Oral morphine-equivalent. ^#^ HT3 receptor antagonists and neurokinin 1 receptor antagonist. Abbreviations: 5-HT3, 5-hydroxytryptamine 3; CSBM, complete spontaneous bowel movement; ECOG PS, Eastern Cooperative Oncology Group performance status; NAL, naldemedine; SBM, spontaneous bowel movement; SD, standard deviation.

**Table 2 cancers-14-02112-t002:** Efficacy in the 2- and 12-week randomized trial.

Endpoints	2 Weeks			12 Weeks		
	1500 mg MgO	0.2 mg NAL	*p* Value	1500 mg MgO	0.2 mg NAL	*p*-Value
	(*n* = 60)	(*n* = 60)		(*n* = 60)	(*n* = 60)	
**Primary endpoint**						
JPAC-QOL Overall	0.5 (0.4)	−0.01 (0.3)	<0.001	0.4 (0.4)	0.03 (0.3)	<0.001
**Secondary endpoints**						
SBM (times/week)	0.3 (1.7)	−0.1 (2.4)	0.3	0.4 (1.8)	0.03 (2.3)	0.4
Stool consistency score	0.6 (1.1)	−0.3 (1.1)	<0.001	0.6 (1.1)	−0.4 (0.9)	<0.001
PAC-SYM Overall	0.6 (0.5)	0.02 (0.3)	<0.001	0.5 (0.5)	0.01 (0.4)	<0.001
ROME IV, *n* (%)	33 (55)	20 (33)	0.02	41 (68)	24 (40)	0.002
CSS	0.3 (0.3)	0.0 (0.3)	<0.001	0.5 (0.4)	−0.02 (0.2)	<0.001
SF-36						
Physical component summary	0 (0)	0.04 (6.6)	1.0	−1.5 (5)	−2.6 (7.2)	0.3
Mental component summary	0 (0)	1.9 (6.1)	0.02	0.2 (5.1)	0.6 (6.2)	0.7
Role component summary	0 (0)	−1.9 (9.7)	0.1	0.9 (7.7)	−1.2 (9.5)	0.2
**Post hoc analyses**						
CSBM (times/week)	−0.9 (1.5)	0 (2.0)	0.01	−0.7 (1.2)	0.2 (2.0)	0.003
JPAC-QOL subscale						
Physical discomfort	0.6 (0.9)	−0.01 (0.6)	<0.001	0.6 (0.8)	0.2 (0.7)	0.01
Psychosocial discomfort	0.6 (0.8)	−0.01 (0.4)	<0.001	0.5 (0.8)	0.1 (0.5)	<0.001
Worries/concerns	0.06 (0.2)	−0.05 (0.4)	0.1	0 (0.5)	−0.1 (0.5)	0.5
Satisfaction	1.0 (0.7)	0.07 (0.5)	<0.001	0.9 (0.8)	0 (0.6)	<0.001
PAC-SYM subscale						
Stool symptoms	0.3 (0.7)	−0.1 (0.6)	<0.001	0.2 (0.7)	−0.1 (0.8)	0.02
Rectal symptoms	0.7 (0.8)	0.1 (0.4)	<0.001	0.6 (0.8)	0.2 (0.6)	<0.001
Abdominal symptoms	0.8 (1.0)	0.1 (0.4)	<0.001	0.7 (0.9)	0.03 (0.6)	<0.001
Mean time to first SBM (h)	4.9 (0.8)	4.9 (1.0)	0.7	4.9 (0.1)	4.9 (0.1)	1.0
Mean time to first CSBM (h)	10.4 (6.4)	6.4 (3.0)	<0.001	10.1 (6.0)	6.4 (3.9)	<0.001
Numerical rating score for pain	−1.4 (1.8)	−1.2 (2.8)	0.7	−1.4 (1.8)	−1.2 (2.8)	0.6

Data are represented as the mean (SD) or number (%). Abbreviations: CSBM, complete spontaneous bowel movement; CSS, constipation scoring system; NAL, naldemedine; JPAC-QOL, Japanese version of Patient Assessment of Constipation Quality of Life; PAC-SYM, Patient Assessment of Constipation Symptoms; SBM, spontaneous bowel movement; SF-36, short form-36.

**Table 3 cancers-14-02112-t003:** Adverse events.

Adverse Events	2 Weeks			12 Weeks		
	1500 mg MgO	0.2 mg NAL	*p*-Value	1500 mg MgO	0.2 mg NAL	*p*-Value
	(*n* = 60)	(*n* = 60)		(*n* = 60)	(*n* = 60)	
Total adverse event	32 (53)	30 (50)		39 (65)	32 (53)	
TRAEs	21 (35)	11 (18)	0.02	31 (52)	16 (27)	0.01
TRAE leading to discontinuation	0	0		0	0	
Serious AEs	0	0		0	0	
Serious TRAEs	0	0		0	0	
Serious TRAE leading to discontinuation	0	0		0	0	
Deaths	0	0		0	0	
TRAEs						
Gastrointestinal disorders SOC						
Abdominal pain	4 (7)	3 (5)		9 (15)	5 (8)	0.3
Diarrhea	4 (7)	5 (8)		4 (7)	6 (10)	
Abdominal distension	1 (0)	0		5 (8)	1 (2)	0.06
Nausea	12 (20)	4 (7)	0.03	20 (33)	7 (12)	0.005

Data are represented as *n* (%). Categorization of adverse drug reactions was based on the Medical Dictionary for Regulatory Activities version 18.0. Abbreviations: NAL, naldemedine; SOC, system organ class; TRAE, treatment-related adverse event.

## Data Availability

The data presented in this study are available on request from the corresponding author.

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
