# Peer review of "Effectiveness of Naldemedine Compared with Magnesium Oxide in Preventing Opioid-Induced Constipation: A Randomized Controlled Trial"

_cancers, 2022, doi:10.3390/cancers14092112_

Round 1

Reviewer 1 Report

This is rigorous designed, well performed study. Congratulations for the authors.

I have only few concerns for the improvement of this manuscript.

  1. According to the baseline characteristics; the text seems to repeat the data within the Table 1. It could be shortened a little bit as the information given in the table is sufficient. There are also repetitions within this paragraph (e.g. “There were 188 no significant differences in baseline background factors between the groups.”) – please delete one of them.
  2. Primary and secondary endpoints paragraph: The sentence “The PAC-QOL subscale (physical discomfort, psychosocial discomfort, and satisfaction) and all mean PAC-SYM subscale (stool symptoms, rectal symptoms, and abdominal symptoms) scores were significantly improved at 2 and 12 weeks.” is not clear in my opinion. Please note if this data pertain to all patients or one of the subgroups. Could this data be presented in the Figure?
  3. Primary and secondary endpoints paragraph; the sentence: “After 2 and 12 weeks of treatment, fewer patients in the NAL group were diagnosed with constipation according to Rome IV criteria” – “fewer” means statistically significant?
  4. Figures – they should have their headings.
  5. Safety outcomes paragraph could be shortened as it repeats the data from the table 3

Author Response

Reviewer 1.

I have only few concerns for the improvement of this manuscript.

  1. According to the baseline characteristics; the text seems to repeat the data within the Table 1. It could be shortened a little bit as the information given in the table is sufficient. There are also repetitions within this paragraph (e.g. “There were 188 no significant differences in baseline background factors between the groups.”) – please delete one of them.

Response: Thank you for your useful suggestion. According to your comments, we shortened the background results and deleted the repetitions.

  1. Primary and secondary endpoints paragraph: The sentence “The PAC-QOL subscale (physical discomfort, psychosocial discomfort, and satisfaction) and all mean PAC-SYM subscale (stool symptoms, rectal symptoms, and abdominal symptoms) scores were significantly improved at 2 and 12 weeks.” is not clear in my opinion. Please note if this data pertain to all patients or one of the subgroups. Could this data be presented in the Figure?

Response: Thank you for your important comments. Sorry for confusing you, the JPAC-QOL and PAC-SYM subscales were summarized in Table 2, and I have added in the text what is clearly stated in Table 2. Moreover, according to your suggestion, we added the change in overall PAC-SYM in figure 2B and JPAC-QOL/PAC-SYM subscale in figure 4A-4D.

  1. Primary and secondary endpoints paragraph; the sentence: “After 2 and 12 weeks of treatment, fewer patients in the NAL group were diagnosed with constipation according to Rome IV criteria” – “fewer” means statistically significant?

Response: Thank you for your helpful suggestion.  As you mention, it was statistically significantly lower in the NAL group compared with MgO group. We clearly stated in the results section that it was statistically significant.

  1. Figures – they should have their headings.

Response: Thank you for your comments. According to your comments, we added headings in figures.

  1. Safety outcomes paragraph could be shortened as it repeats the data from the table 3.

Response: Thank you for your useful suggestion. According to your comments, we shortened the safety outcomes paragraph.

Reviewer 2 Report

The current paper compares effectiveness of Nalmedine compared to Magnesium Oxide in preventing opioid induced constipation.  Primary endpoint was the average change in the Japanese version of patient assessment of constipation after 2 weeks and 12 weeks. The results showed significant better outcome in the Nalmedine group regarding the primary endpoint and the side effects.  There was no difference in the spontaneous bowel movements. However, complete spontaneous bowel movements were higher in the Nalmedine group.

The paper as well written and deals with an important clinical problem.  The study design is appropriate and the results are presented in an appropriate manner. 

One question remains regarding this study: The use of magnesium oxide as comparatortor should be explained in many countries osmotic laxatives like Macrogol is used for treating and as prophylaxis of opioid-induced constipation.   A short explanation why magnesium oxide is used as comparator could be included in the paper.

Author Response

Reviewer 2

The current paper compares effectiveness of naldemedine compared to Magnesium Oxide in preventing opioid induced constipation.  Primary endpoint was the average change in the Japanese version of patient assessment of constipation after 2 weeks and 12 weeks. The results showed significant better outcome in the naldemedine group regarding the primary endpoint and the side effects.  There was no difference in the spontaneous bowel movements. However, complete spontaneous bowel movements were higher in the naldemedine group. The paper as well written and deals with an important clinical problem.  The study design is appropriate and the results are presented in an appropriate manner.

One question remains regarding this study: The use of magnesium oxide as comparator should be explained in many countries osmotic laxatives like Macrogol is used for treating and as prophylaxis of opioid-induced constipation.  A short explanation why magnesium oxide is used as comparator could be included in the paper.

Response: Thank you very much for your helpful suggestion. According to your suggestion, we added a short explanation why magnesium oxide was used as comparator in introduction section. 

Reviewer 3 Report

Oncological patients treated with opioids are significantly affected by constipation, even more stubborn if immobile or with coexisting gastrointestinal pathologies. This study has compared the effectiveness of magnesium oxide (MgO) with that of naldemedine (NAL) in preventing Opioid-induced constipation. Patients treated with naldemedine compared to magnesium oxide, (the latter a very widespread osmolar laxative) led to a better therapeutic response. But it should be remembered that the patient cohort must be selected by screening patients at risk of perforation in the intestine such as diverticulitis and dolichosigma.

It is an original argument as only this research group has already marked a clinical trial(UMIN000031891) from which for retrospective study it has deepened its clinical and quality of life aspects.

Adverse, sometimes fatal, effects of naldemedine in patients with diverticulitis(minor revision) which have been neglected perhaps due to being administered to terminally ill cancer patients should be considered.

Nevertheless, the study conducted demonstrates the arguments presented
with methodological rigor and with an appropriate statistical distribution
and supported by appropriate and updated references.

Author Response

Reviewer 3

Oncological patients treated with opioids are significantly affected by constipation, even more stubborn if immobile or with coexisting gastrointestinal pathologies. This study has compared the effectiveness of magnesium oxide (MgO) with that of naldemedine (NAL) in preventing Opioid-induced constipation. Patients treated with naldemedine compared to magnesium oxide, (the latter a very widespread osmolar laxative) led to a better therapeutic response. But it should be remembered that the patient cohort must be selected by screening patients at risk of perforation in the intestine such as diverticulitis and dolichosigma. It is an original argument as only this research group has already marked a clinical trial(UMIN000031891) from which for retrospective study it has deepened its clinical and quality of life aspects. Adverse, sometimes fatal, effects of naldemedine in patients with diverticulitis(minor revision) which have been neglected perhaps due to being administered to terminally ill cancer patients should be considered. Nevertheless, the study conducted demonstrates the arguments presented with methodological rigor and with an appropriate statistical distribution

and supported by appropriate and updated references.

Response: As you mentioned, we believe it is very important to target patients at risk for diverticulitis and other intestinal perforations. In this study, we excluded patients at risk of bowel perforation, taking into account the safety of terminally ill patients (Supplementary table 1). Since the safety of naldemedine in this study was ensured, we believe that future studies should be conducted in patients with diverticulitis or dolichosigma. Thanks for your suggestion, and I will try to find out more about the safety of naldemedine in diverticulitis or dolichosigma.

Reviewer 4 Report

This paper describes the use of naldemedine as an alternative medication to prevent opioid-induced constipation in cancer patients.

This well designed, prospective, randomized trial speaks for a better long-term QoL in patients taking naldemedine, as compared to ones taking magnesium oxide. The study brings well supported new information on cancer patients care, which may be useful in clinical practice.

-The study addresses a problem of constipation in cancer patients using opioids.
The advantage of oral medication with naldemedine (NAL)  0.2 mg 3x/d oral over MgO 500 mg 3x/d oral was investigated in a randomized trial in 120 patients over 12 weeks. The study proved a better QoL in NAL patients using opioids for analgesia

-This well documented and well written study may help to further improve a widely-used analgetic treatment in cancer patients, thus may influence our daily clinical practice.

-There are already a few papers on naldemedine treatment to prevent opioid-induced constipation.
This study delivers further argument to improve well known negative consequences of treatment with opioids in cancer patients.

-The Authors use a well designed study and proper statiscics.
The paper is not a break-through, but is a source of well supported data to use naldemedine insead of MgO.

-The arguments are consistent with our to-date knowledge and directly address the topic.

-The references are appropriate.

I recommend this article to be published in Cancers journal.

Author Response

Reviewer 4

This paper describes the use of naldemedine as an alternative medication to prevent opioid-induced constipation in cancer patients. This well designed, prospective, randomized trial speaks for a better long-term QoL in patients taking naldemedine, as compared to ones taking magnesium oxide. The study brings well supported new information on cancer patients care, which may be useful in clinical practice.

-The study addresses a problem of constipation in cancer patients using opioids.

The advantage of oral medication with naldemedine (NAL) 0.2 mg 3x/d oral over MgO 500 mg 3x/d oral was investigated in a randomized trial in 120 patients over 12 weeks. The study proved a better QoL in NAL patients using opioids for analgesia

-This well documented and well written study may help to further improve a widely-used analgetic treatment in cancer patients, thus may influence our daily clinical practice.

-There are already a few papers on naldemedine treatment to prevent opioid-induced constipation.

This study delivers further argument to improve well known negative consequences of treatment with opioids in cancer patients.

-The Authors use a well designed study and proper statiscics.

The paper is not a break-through, but is a source of well supported data to use naldemedine insead of MgO.

-The arguments are consistent with our to-date knowledge and directly address the topic.

-The references are appropriate.

I recommend this article to be published in Cancers journal.

Response: Thank you very much for your helpful comments. We greatly appreciate you spending your valuable time on our dissertation review.